# Acquisition of Streptomycin Resistance by Oxidative Stress Induced by Hydrogen Peroxide in Radiation-Resistant Bacterium *Deinococcus geothermalis*

**DOI:** 10.3390/ijms23179764

**Published:** 2022-08-28

**Authors:** Chanjae Lee, Qianying Ye, Eunjung Shin, Tian Ting, Sung-Jae Lee

**Affiliations:** Department of Biology, Kyung Hee University, Seoul 02447, Korea

**Keywords:** *Deinococcus geothermalis*, chronic oxidative stress, insertion sequence, phenotypic variation, ribosomal protein S12 (RpsL), streptomycin-dependent (SmD) and -resistant (SmR), transposition

## Abstract

Streptomycin is used primarily to treat bacterial infections, including brucellosis, plague, and tuberculosis. Streptomycin resistance easily develops in numerous bacteria through the inhibition of antibiotic transfer, the production of aminoglycoside-modifying enzymes, or mutations in ribosomal components with clinical doses of streptomycin treatment. (1) Background: A transposable insertion sequence is one of the mutation agents in bacterial genomes under oxidative stress. (2) Methods: In the radiation-resistant bacterium *Deinococcus geothermalis* subjected to chronic oxidative stress induced by 20 mM hydrogen peroxide, active transposition of an insertion sequence element and several point mutations in three streptomycin resistance (SmR)-related genes (*rsmG*, *rpsL*, and *mthA*) were identified. (3) Results: IS*Dge6* of the IS*5* family integrated into the *rsmG* gene (*dgeo*_2335), called S*rsmG*, encodes a ribosomal guanosine methyltransferase resulting in streptomycin resistance. In the case of *dgeo*_2840-disrupted mutant strains (S1 and S2), growth inhibition under antibiotic-free conditions was recovered with increased growth yields in the presence of 50 µg/mL streptomycin due to a streptomycin-dependent (SmD) mutation. These mutants have a predicted proline-to-leucine substitution at the 91st residue of ribosomal protein S12 in the decoding center. (4) Conclusions: Our findings show that the active transposition of a unique IS element under oxidative stress conditions conferred antibiotic resistance through the disruption of *rsmG*. Furthermore, chronic oxidative stress induced by hydrogen peroxide also induced streptomycin resistance caused by point and frameshift mutations of streptomycin-interacting residues such as K43, K88, and P91 in RpsL and four genes for streptomycin resistance.

## 1. Introduction

Despite the development and widespread use of antibiotics, bacteria have the upper hand against humans. Currently, many sectors worldwide, including healthcare and commercial, face a serious crisis in the effective treatment of bacterial infection, reaching the point where cocktails of different antibiotics are needed against infectious bacteria. Because of the inherently enormous selective pressure of antibiotics, bacteria readily acquire antibiotic resistance, which can be attributed to antibiotic uptake inhibition, a change in the target or mechanism of action, or direct degradation, particularly of ribosome-targeting antibiotics [1,2,3].

The bacterial ribosome is a major target of antibiotics, as its inhibition results in the cessation of protein synthesis. Ribosome-targeting antibiotics, such as the aminoglycoside antibiotic streptomycin, which was the first discovered, block protein synthesis by binding to functionally important regions of bacterial rRNA of the small 30S ribosomal subunit [4,5]. The streptomycin-binding sites comprise 16S rRNA helices 11, 18, 27, and 44 and ribosomal protein S12. Streptomycin induces significant local distortion of 16S rRNA, including the crucial bases A1492 and A1493 of helix 44, which directly participate in codon recognition, thereby disrupting protein synthesis [5].

Streptomycin was first used in 1943, and the first streptomycin-resistant strain appeared in 1946 [6]. Since the initial discovery, streptomycin (Sm) resistance mechanisms have been constantly elucidated. Before 1950, Paine and Finland reported Sm-pseudo-dependent (SmP), Sm-dependent (SmD), and Sm-resistant (SmR) phenotypes with streptomycin in a concentration range of 0–50,000 µg/mL in four bacteria [7,8]. SmD mutants grow at a specific threshold concentration of streptomycin, for example, 50 and 12.5 µg/mL for *Staphylococcus aureus* and *Escherichia coli*, respectively. Further, Ravin and Mishra reported three classes of streptomycin-resistant bacteria depending on the concentration, termed very low resistance, low resistance, and high resistance, that occurred spontaneously and induced *Streptococci* and *Pneumococci* mutants [9]. In the 1970s, Gupta and Schlessinger reported physiological studies of SmD mutants using transcription and translation rates and mRNA degradation [10]. According to modern research, SmR mutant strains have two phenotypic differences: (1) a low-level resistant phenotype derived from mutations in two genes, *rsmG* and *mthA*, which encode a methyltransferase of 16S rRNA and a methylthioadenosine nucleosidase in the S-adenosylmethionine (SAM) recycling pathway, respectively; and (2) a high-level resistant phenotype caused by the mutation of *rpsL*, which encodes a ribosomal S12 component [11,12]. Antibiotic resistance is often conferred by the addition of a methyl group at the antibiotic-binding site in the rRNA region through its own specific methyltransferase enzyme. RsmG is an AdoMet-dependent methyltransferase responsible for the synthesis of m^7^G527 in the 530 loop of bacterial 16S rRNA [13]. The 530 loop region is universally conserved and plays a key role in ribosomal accuracy; it is also a target of streptomycin binding, especially at nucleotides G526 and G527 [14]. Thus, when conserved *rsm*G is mutated, bacteria such as *Escherichia coli*, *Mycobacterium tuberculosis*, *Bacillus subtilis*, and *Streptomyces coelicolor* exhibit low-level resistance to 100 µg/mL streptomycin as well as either an increase or decrease in mRNA misreading [13,15,16,17]. In contrast to the low-level resistant phenotype, the high-level SmR phenotype, which is caused by a mutation in the ribosomal component RpsL, a ribosomal S12 protein of *B. subtilis* mutant strains, revealed a resistance phenotype exceeding a streptomycin concentration of 10,000 µg/mL [16]. Further, the accumulation of mutations in *rpsL*, *mthA*, and *rpoB* confers a greater SmR phenotype and increases antibiotic production in *Bacillus* [12,16]. Similarly, in *Streptomyces* species, mutations in two ribosomal components, RpsL and RsmG, resulted in increased streptomycin resistance and antibiotic production [18]. Additionally, in *Salmonella enterica*, high-level resistance to streptomycin was associated with increased expression of *aadA*, which encodes an aminoglycoside adenylyltransferase, and is controlled by a guanosine penta/tetraphosphate [(p)ppGpp]. The deletion of *zunA*, a gene that encodes a periplasmic zinc-binding protein, increased the transcriptional level of *aadA* [3,19,20]. The technology used for these studies was ribosome engineering, a process used for drug screening that has several advantages over simple selection.

Similar to other bacteria, SmR mutants of *Thermus*
*thermophilus* were spontaneously selected through serial cultivation with gradual streptomycin treatment. Most mutants revealed point mutations in streptomycin-interacting residues of ribosomal protein S12 [11,21]. Further, investigation of the mutation positions in RpsL and their effects on streptomycin interaction showed that these amino acid substitution mutants exhibited different phenotypes, such as SmR at P41S, K42R, K87R/E, and P90A/C/G, SmP at R85C/H, and SmD at P90E/M/R/W, in the S12 protein. Additionally, a mutation in the target gene *rsmG* through transposition of the insertion sequence (IS) element IS*Tth7* resulted in streptomycin resistance in *T. thermophilus* HB8 [22]. This IS element transposition was selected from a large collection of spontaneous SmR mutants.

While searching for a new biomarker to track the potential transposition phenomenon of insertion sequence (IS) elements in the radiation-resistant bacterium *Deinococcus geothermalis*, it was discovered that an IS element was integrated into the *rsmG* gene in a strain that showed streptomycin resistance [23,24]. Further, some SmR mutations caused by point and frameshift mutations under chronic oxidative stress conditions resulted in two wild-type *dps* gene (*dgeo*_0257 and *dgeo*_0281)-disrupted mutants and a putative LysR family regulator (*dgeo*_2840)-disrupted mutant [25,26,27]. In this study, we characterized the transposed IS element, transposition and mutation loci, and the mode of transposition with alternative SmR screening of various mutations in *rsmG*, *rpsL*, *mthA*, and other potential target genes involved in antibiotic uptake and modification. As a result, we elucidated the SmR phenotype caused by the oxidative stress response to H_2_O_2_ treatment in an extreme gamma radiation-resistant bacterium. The phenotypic SmR frequency was 1.36 × 10^−7^ in the wild-type strain, which is slightly higher than non-pigment production by IS transposition under H_2_O_2_ treatment [27]. Additionally, these findings contribute to a better understanding of the mechanism of streptomycin resistance and ribosomal structure resolution, in particular, the small ribosomal subunit structure for streptomycin binding.

## 2. Results

### 2.1. Selection of Streptomycin Resistance as a New IS Transposition Biomarker

Two wild-type *dps*-disrupted mutant strains, Δ*dgeo*_0257 and Δ*dgeo*_0281, and a LysR family regulator gene Δ*dgeo*_2840-disrupted mutant strain were grown on TGY plates in the absence of streptomycin. From strains chronically exposed to oxidative stress using 20 mM H_2_O_2_ in four overnight continuous cultures, several SmR clones were selected (Figure 1A). Both S1 and S2 mutants of Δ*dgeo*_2840 exhibited possible Sm-dependent (SmD) phenotypes, as shown in the red box. Four clones from the wild type, one clone from each *dps*-disrupted mutant, and three clones from the Δ*dgeo*_2840 mutant strain were selected. We then determined the MIC levels of these selected clones and sorted them based on the level of streptomycin resistance (Figure 1B). The MIC values and mutations in SmR-related genes are summarized in Table 1. Unlike the known SmR levels of other bacteria, MIC levels were parsed into either 10,000 µg/mL (low resistance) or greater than 20,000 µg/mL (high resistance) in *D. geothermalis*. The low-resistance clones were S1 of the Δ*dgeo*_0257 mutant and a *rsmG*-disrupted mutant (S*rsmG*) caused by IS transposition in a Δ*dgeo*_2840-disrupted mutant, whereas the high-resistance clones were SmR mutants of the wild type, S1 of the Δ*dgeo*_0281 mutant strain, and S1 and S2 of the Δ*dgeo*_2840-disrupted mutant.

### 2.2. Growth Effects of Streptomycin

The isolated streptomycin-resistant strains from the wild type, Δ*dgeo*_0257, and Δ*dgeo*_0281 exhibited less growth than parent strains without antibiotic treatment. However, when 50 µg/mL streptomycin was present on TGY medium, the resistant strains revealed different growth ratios; that is, Δ*dgeo*_0257 S1 and Δ*dgeo*_0281 S1 mutants exhibited slightly better growth than wild-type-originated resistant strains (Appendix A). Thus, the isolated antibiotic-resistant strains may have different mutations in streptomycin resistance-related genes.

When we attempted to cultivate SmD mutants on TGY medium overnight, we found that the Δ*dgeo*_2840 S1 and S2 strains did not grow (Figure 2A). However, in the presence of either 50 or 100 µg/mL streptomycin, both mutant strains grew well (Figure 2B), and it was not until using streptomycin concentrations of 500 or 1000 µg/mL that the growth of both mutant strains was inhibited. Interestingly, based on measurements of MIC values using the disc diffusion method, we discovered that the Δ*dgeo*_2840 S1 and S2 mutants grew well in a streptomycin concentration-dependent manner (Figure 2B). In fact, because the Δ*dgeo*_2840 S1 and S2 mutants of *D. geothermalis* exhibited a similar phenotype to the SmD phenotype of *T. thermophilus* with P90E/M/R/W of RpsL (S12 ribosomal protein) [21], it is possible that both SmD mutants of *D. geothermalis* have a mutation at the P91 residue.

### 2.3. Detection of IS Transposition

We discovered an SmR mutant as a novel selectable biomarker for IS element transposition that was isolated from the Δ*dgeo_2840* strain (Figure 3). The selected mutant S*rsmG* showed a longer PCR fragment with *dgeo*_2335 only (Figure 3, in lane 9 of *dgeo*_2335). The IS*Dge6* element was integrated into the *rsm*G gene at the 58th nucleotide (Figure 4A). Further, we determined that the element is a member of the IS*5* family, containing a DDE-motif transposase that is 335 amino acids in length, as well as a conserved terminal inverted repeat (TIR) sequence of ‘AGACCtGCTGCAAAcaAGGGGC’, and a direct repeat (DR) sequence of ‘TCA’. In our previous study, we showed that IS*Dge6* actively transposed into and disrupted phytoene desaturase, an enzyme required for carotenoid biosynthesis, in the Δ*dgeo_*2840 mutant strain [27]. The other SmR mutations revealed reddish colors in culture, indicating functional carotenoid synthesis and thus the absence of IS integration. Based on these findings, the Δ*dgeo_*2840 strain-specific IS*5* family element IS*Dge6* transposed into *rsmG* and was detected through SmR selection by oxidative stress.

### 2.4. Detection of Point and Frameshift Mutations in Ribosome-Related Genes

To detect point mutations, four target genes-related to known SmR-involved genes, *rsmG* (*dgeo*_2335), *rpsL* (*dgeo*_1873), and two *mthA* (*dgeo*_0447 and *dgeo*_0776), were selected for DNA sequence analysis. All genes were sequenced, and point mutations were identified in the SmR strains. A summary of all mutations in SmR mutants is shown in Table 1 and Figure 4. In *rpsL* of the four SmR mutants of the wild type, a point mutation was found at the 272nd nucleotide in which ‘C’ was replaced with ‘A’, resulting in a predicted amino acid change from proline to histidine at the 91st residue. Mutant S2 of the wild type had an additional mutation at *dgeo*_0776, which was a nucleotide deletion at the 84th nucleotide, resulting in a frameshift mutation. The Δ*dgeo*_0257 S1 strain exhibited a mutation at the 262nd nucleotide with an A-to-G substitution in *rpsL*, resulting in a predicted lysine-to-glutamic acid change at the 88th amino acid residue and an additional deletion mutation at *dgeo*_2335. The Δ*dgeo*_0281 S1 strain had the same mutations in *rpsL* as the wild type, with two mutations in both *mthA* genes: one was a G-to-C substitution at position 102 in *dgeo*_0776 but with no predicted amino acid change, and the other was a nucleotide deletion at the 627th nucleotide, resulting in a frameshift mutation in *dgeo*_0447. Both Δ*dgeo*_2840 S1 and S2 strains revealed a point mutation in *rpsL* compared to the wild-type SmR mutant. However, at the 272nd nucleotide, we found a C-to-T substitution, with a predicted amino acid change from proline to leucine at the 91st residue and a phenotypic change to streptomycin resistance. Nevertheless, Δ*dgeo*_2840 S1 and S2 strains exhibited somewhat different growth patterns; at the moment, we have not found any additional mutations in the selected Sm resistance-related genes. Importantly, from the mutants identified in our analysis, we found that the K88E and P91H mutations of RpsL were responsible for the SmR phenotype, and the P91L mutation of RpsL caused the SmD phenotype.

### 2.5. In Silico Analysis of Protein Biosynthesis Control

The SmR phenotype of the RpsL protein had amino acid substitutions of P91H in the wild-type and Δ*dgeo*_0281 mutants and K88E in Δ*dgeo*_0257. The SmD phenotype of RpsL was derived from P91L in the Δ*dgeo*_2840 mutant. The interaction model between RpsL and streptomycin was derived from the ribosome structure of the *T. thermophilus* small 30S subunit obtained from the Protein Data Bank (PDB) with accession number 4DV7 (Figure 5 and Appendix A). Examination of the ribosomal structure using Mol* Viewer [28] showed the decoding center for protein biosynthesis in the ‘A’ site of the ribosome among the ribosomal proteins S4, S12, and 16S rRNA with streptomycin. Additionally, we illustrated that amino acid residues K43 and K88 of protein S12 (based on the amino acid alignment shown in Appendix A) are involved in streptomycin interaction, and specific nucleotides of helices 18, 27, and 44 of the 16S rRNA are involved in codon recognition at the decoding center of the small ribosomal subunit. The interaction of the aminoglycoside antibiotic streptomycin with this decoding center results in the inhibition of protein biosynthesis and translation arrest. However, mutations in this area, such as K43, K88, and P91 of S12, inhibit streptomycin interaction, thereby resulting in dramatically increased streptomycin MIC values. Carr et al. predicted that the bulkier hydrophobic amino acid leucine imposes a steric and/or charge-related effect on 16S rRNA that favors the open conformation [29]. We predict that this distance change affects the control of the translational ratio, especially in the case of P91L in Δ*dgeo*_2840. In other words, all translation is blocked in this area, with the arrest of tRNA and elongation factors at the locus. However, in the presence of streptomycin, this distance between the S12 protein and 16S rRNA is closed, enabling continuous translation by tRNA and elongation factors. Therefore, the point mutation P91L inhibited streptomycin interaction in the RpsL structure for the recognition of EF-Tu and tRNA synthase, resulting in the protection of protein synthesis.

### 2.6. qRT-PCR of Translational Factors for SmD Cell Growth

To determine translational ratio control in SmD cell growth, we selected two EF-Tu genes and an antibiotic chemical modifier that enhances their expression. As *D. geothermalis* lacks genes corresponding to *aadA* for chemical modification and *strA*/*B* for degradation, instead, we chose a putative aminoglycoside 3-N-acetyltransferase gene (*dgeo*_1202) as an antibiotic modifier and tested the level of gene expression [18]. We also examined the expression levels of two EF-Tu (*dgeo*_0646 and *dgeo*_1869) because the translational events were limited by the EF-Tu factor as an A-site decoding reaction with GTP hydrolysis and signal relay with ribosomal protein S12 [30]. The expression levels of EF-Tu factors and an antibiotic modifier were induced in a growth phase-dependent manner, except for the *dgeo*_2840-disrupted mutant strain. Both EF-Tu factors and *dgeo*_1202 were significantly induced more than 2-fold in the early growth phase at OD_600_ of 2.0 compared to the wild-type strain. Both Δ*dgeo*_2840 S1 and S2 SmD mutant strains were more significantly induced than their parent strain (Figure 6). Thus, in SmD mutants of the Δ*dgeo*_2840 strain, both EF-Tu and putative *aadA* are significantly upregulated in the 2.0 and 4.0 OD_600_ growth phases and somehow affect their protein synthesis.

### 2.7. Detection of Antibiotic Uptake and Modifying Genes

There are several additional genes involved in the mechanism of antibiotic resistance, including those involved in transport, enzyme modification, and degradation [3]. Thus, we selected the following three transporter genes and a ribosomal protein for DNA sequencing and mutation analysis: *dgeo*_1583 (*trkH*) for potassium uptake, *dgeo*_0915 (*nuoG*) for antibiotic uptake, *dgeo*_0534 (*znuA*) for zinc uptake, and *dgeo*_1841 for the S4 protein. We found no differences in the length of the PCR fragments of the three transporter genes or the S4 gene (Appendix A). Additionally, to identify point mutations in these three transporter genes, the target genes were PCR-amplified, and DNA sequences were analyzed. We detected several point mutations in *nuoG* in S3 of the wild type and S1 of the Δ*dgeo*_0281 mutant strain (Table 1). The *nuoG* gene of the Δ*dgeo*_0281 S1 strain exhibited three point mutations, but amino acid substitution only occurred at the 1462nd residue, resulting in a change from Gly to Arg exchange.

DNA sequencing analysis of the PCR products of four 16S rRNAs was confirmed, and no mutations were identified (Appendix A). In particular, the S1 mutant of the Δ*dgeo*_0281 strain exhibited a MIC of streptomycin exceeding 50,000 µg/mL through the accumulation of point and frameshift mutations.

## 3. Discussion

The emergence of the SmR phenotype in bacteria readily occurred to a large extent because of the wide commercial use of streptomycin as medical treatment to control microbial pathogens and as an animal feed additive for growth promotion. Streptomycin has improved in efficiency for over 50 years, although recently, the antibiotic has been substituted by various alternatives, such as prebiotics, probiotics, synbiotics, and antimicrobial peptides [31,32]. In fact, the three types of streptomycin phenotypes in bacteria, namely, SmR, SmP, and SmD, are the result of substituted amino acid residues at the site of streptomycin interaction [11]. Interestingly, a well-known probiotic, *Lactobacillus* species, has extremely high MIC levels after low-level exposure to streptomycin. The strain has an *rpsL* gene mutation, and the accumulation of mutations through antibiotic pressure resulted in high-level resistance exceeding 131,000 µg/mL streptomycin after 25 days. This SmR strain quickly lost the SmR phenotype without antibiotic pressure but maintained resistance to 2040 µg/mL streptomycin after 35 days [33]. Similarly, *E. coli* isolated from an environmental sea product revealed resistance to more than 1024 µg/mL streptomycin [34]. Laboratory experiments in evolution have also shown a high level of antibiotic resistance stemming from antibiotic exposure in different bacteria [3,35,36]. In fact, various bacteria readily gained streptomycin resistance with a wide range of MICs. Since the analysis of *strA* (which encodes an aminoglycoside phosphotransferase) point mutations in *E. coli* in 1972, there has been significant research into SmR mechanisms, including the analysis of various point mutations and structural resolutions of conformational changes to ribosomal proteins in *E. coli*, *Salmonella* Typhimurium, and *T. thermophilus* [4,37,38,39].

Following the understanding of the translational machinery and ribosomal structure, as well as the development of different protein mutation assays, the mechanisms underlying resistance to aminoglycoside antibiotics have been defined for the past half-century. Two helices of 16S rRNA, h18 (525th and 526th nucleotides) and h27 (912th and 913th nucleotides) of the central pseudoknot area of 16S rRNA, interact with two loop regions, namely, K43K44P45 and K88 of ribosomal protein S12, as shown in Appendix A [40]. A loop of the S12 protein interacts with the tRNA anticodon stem-loop structure at the ‘A’ site and stabilizes the bases of 16S rRNA that are involved in the recognition of codon–anticodon base pairing [41]. Streptomycin binds between the 16S rRNA and S12 interaction area, resulting in translational misreading [37]. A deficiency in N-7 methylation of G527 of 16S rRNA by the methyltransferase RsmG produces low-level streptomycin resistance. Point and frameshift mutations in the 252 loop region of *rsmG* inactivate 16S rRNA methyltransferase activity, resulting in a moderate SmR phenotype in the range of several 100 µg/mL. In particular, the K43, K44, and K88 lysine residues of ribosomal protein S12 (RpsL) are directly involved in this interaction with streptomycin [4]. Further, the tertiary interaction between the 16S rRNA helices h44 and h45 is destabilized by streptomycin interaction [5]. Thus, if this region is mutated, the mutant strains reveal a higher MIC level of resistance exceeding 32,000 µg/mL [13]. Additionally, *mthA* is involved in the low-level SmR phenotype [42].

In this study, we selected SmR and SmD mutants derived in response to chronic oxidative stress using a relatively low concentration of hydrogen peroxide (20 mM) from the wild type, two Dps-disrupted strains, and a LysR family regulator-disrupted strain as parent cells of *D. geothermalis* (Table 1). The *rpsL* point mutation P91H resulted in a MIC value of almost 50,000 µg/mL, and when point and frameshift mutations were accumulated with mutations in *mthA*, the value increased to more than 50,000 µg/mL in the Δ*dgeo*_0281-originated SmR mutant. A Dps gene-disrupted mutant revealed slightly different MIC values of 20,000 µg/mL because of different point mutations and their accumulation in the case of the Δ*dgeo*_0257 SmR mutant. The point mutation K88E in the Δ*dgeo*_0257 SmR mutant was related to the streptomycin interaction site in *rpsL* of *E. coli* (Figure 5 and Appendix A) [43]. The LysR family regulator-disrupted mutant included P91L in RpsL and had two SmD mutants. These P91L mutations in RpsL were already identified in *Salmonella typhimurium* [41]. In the absence of streptomycin, cell growth was stalled, whereas cell growth restarted following the addition of streptomycin at a concentration above 50 µg/mL. This growth pattern of SmD was reported during early streptomycin commercial usage at the end of the 1940s [7,8]. Both mutants exhibited a similar streptomycin MIC level of 50,000 µg/mL. Evaluation of cell growth measurements with streptomycin showed that Δ*dgeo*_2840 S2 grew better than the Δ*dgeo*_2840 S1 mutant. The complement strain C2840 exhibits different streptomycin-resistant MIC values after chronic hydrogen peroxide treatment with 20 mM and has mutations in K43, K88, and P91 on RpsL compared to parent strains (Appendix A). The complement strain shows a recovered growth pattern and hydrogen peroxide sensitivity (Appendix A). Based on our findings, we hypothesize that streptomycin interacts with the P91L residue and causes translation to restart and resume protein production, and consequently, cells are released from growth arrest. Other streptomycin-interacting residues, such as K43 and K88 of the streptomycin interaction sites in RpsL, thereby overcome translational inhibition and effectively restart protein synthesis.

In the case of the thermophilic bacterium *T. thermophilus*, the K43 mutation leads to an SmR phenotype in the range of 500–2000 µg/mL, and several P91 (which corresponds to the amino acid sequence in Appendix A) mutations revealed an SmD phenotype with substitutions to arginine, glutamate, methionine, and tryptophan [21]. However, the P91A, P91C, and P91G mutants reveal an SmR phenotype. Thus, the charge and hydrophobic properties of the side chain of the amino acid residues in RpsL affect the growth phenotype of the mutant strains.

Additionally, gene disruption by the transposition of an IS element affects streptomycin resistance in bacteria. In *E. coli*, high glucose concentrations affect IS*1* transposition into *rpsL* [44], and in *T. thermophilus* HB8, the spontaneous SmR strain has IS*Tth7* integrated into *rsmG* [22], resulting in streptomycin resistance. IS*Tth7* is a member of the IS*427* group within the IS*5* family. IS*Tth7* produces a variable direct repeat sequence of 9 bp and has a transposase containing a DDE motif and a second ORF, termed OrfA [22]. In previous experiments on IS transposition, we discovered that the transposition of IS*Dge**6*, IS*Dge**7*, and IS*Dge**11* into *dgeo*_0524 occurred in response to oxidative stress, which encodes a phytoene desaturase for carotenoid biosynthesis, in the Δ*dgeo*_2840 mutant, Δ*dgeo*_0257 mutant, and wild type, respectively [26,27]. In this study, we also identified IS transposition into *rsmG* in the Δ*dgeo*_2840 mutant strain of *D. geothermalis*. IS*Dge6* of the IS*5* family integrated at the 58th nucleotide of the *rsmG* gene, resulting in an SmR phenotype with a MIC value of more than 10,000 µg/mL streptomycin (Figure 1B and Table 1).

The SmD mutants grew well with the addition of streptomycin in the culture medium. Analysis of measurements of the rates of transcription and translation and mRNA degradation indicated that SmD mutants have translation rates proportional to growth rates through threshold levels of streptomycin [10]. In this study, we measured the expression levels of several translational factors, including two EF-Tu and AadA, because both factors are involved in correct codon–anticodon interactions resulting in enhanced translation yield [41]. The expression levels of these genes correlated well with the LysR family regulator-disrupted mutant strain (Figure 6). Further, the SmR mutants revealed IS element transposition into the SmR-related *rsmG* and point mutations in a specific ribosomal protein (S12) and a 16S rRNA chemical modifier enzyme in the radiation-resistant bacterium *D. geothermalis*. Therefore, the SmR and SmD phenotypes caused by chronic oxidative stress are similar to the spontaneous mutation caused by direct low-level exposure to streptomycin.

Furthermore, this observation suggests that microbial pathogens, including *M. tuberculosis*, *Staphylococcus aureus*, *S.* Typhimurium, and *E. coli*, undergo phenotypic variation to develop antibiotic resistance through IS transposition in response to oxidative stress, particularly to ROS accumulation. Recent reports supported the possibility that ROS accumulate in response to antibiotic treatment and contribute to the killing mechanism in bacteria [45,46,47]. Therefore, this is an important insight for further research on antibiotic resistance acquisition by IS transposition in pathogens.

## 4. Materials and Methods

### 4.1. Bacterial Strains and Culture Conditions

Based on the wild-type strain *D. geothermalis* DSM 11300^T^, which was provided by the Korean Agricultural Culture Collection (KACC, http://genebank.rda.go.kr/, 15 December 2008), we constructed target gene knock-out mutants of *dgeo_*0257 and *dgeo*_0281 for both DNA-protecting proteins from starved cells (Dps) and *dgeo*_2840 (LysR family transcriptional regulator gene) by homologous recombination using a pKATaph plasmid containing a kanamycin resistance cassette, as described in previous studies [26,27,48]. The complement strain for Δ*dgeo*_2840 was constructed using a shuttle vector pRADgro including a *dgeo*_2840 gene [45]. The complex culture medium for the *Deinococcus* wild type, mutant strains, and the complement strain was the commonly used TGY medium, containing 1% tryptone (Duchefa Biochemie, Haarlem, The Netherlands), 0.1% glucose (Sigma-Aldrich, St. Louis, USA), and 0.5% yeast extract (Duchefa Biochemie), and the strains were incubated overnight at 48 °C. SmR strains were selected in TGY medium containing 50 µg/mL streptomycin (Streptomycin sulfate, Sigma-Aldrich). Bacterial growth was measured by optical density (OD) with a wavelength of 600 nm (OD_600_) in TGY broth medium with or without streptomycin at different concentrations.

### 4.2. Chronic Oxidative Stress

For chronic oxidative stress conditions, the wild type, mutant strains, and the complement strain grown to an OD_600_ of 4.0 were exposed to 20 mM H_2_O_2_ (Hydrogen peroxide, Wako, Japan) in TGY media overnight. Every 24 h, the overnight cultured cells were diluted to an OD_600_ of 4.0 with fresh TGY medium containing 20 mM H_2_O_2_ and incubated for 4 days. The tested strains grew well under H_2_O_2_ conditions without growth delay. After the 4th overnight culture, cells were diluted to 10^−5^ and spread on a TGY agar plate containing 50 µg/mL streptomycin. The plates were then incubated at 48 °C for 2 days, and colonies were selected as SmR strains.

### 4.3. Determination of Streptomycin Minimum Inhibitory Concentration (MIC)

MIC values were measured for direct bacterial cultures grown on TGY agar plates with different concentrations of streptomycin using the disc diffusion method. The MIC value was determined as the maximal concentration of streptomycin until the appearance of an inhibition zone around the disc. The conditions used for bacterial incubation were identical to culture conditions.

### 4.4. Growth Curve Determination

To analyze the growth patterns of the wild type and mutants, all strains were grown overnight in TGY or TGY containing 50 µg/mL streptomycin. After overnight culture, strains were diluted to an OD_600_ of 2.0, and growth pattern measurements were started from an OD_600_ of 0.06 of the main culture in 50 mL of TGY medium. OD_600_ was measured hourly using a Bio-Rad spectrophotometer (Hercules, CA, USA). For SmD strains, growth patterns were measured on TGY media containing 50 µg/mL streptomycin 3 h later. To determine the growth patterns of Δ*dgeo*_2840 and the complement strain, Δ*dgeo*_2840 containing the empty pRADgro vector and the complement strain were grown on TGY media containing 3 µg/mL chloramphenicol as a selection marker. It was observed that 100 µg/mL streptomycin similarly affected cell growth; however, 500 and 1000 µg/mL streptomycin exhibited the inhibition of cell growth.

### 4.5. Detection of Transposition Loci

To detect transposition loci, we used primer sets encompassing our four streptomycin resistance-related target genes, namely, *dgeo*_1873 (*rpsL*), *dgeo*_2335 (*rsmG*), *dgeo*_0776, and *dgeo*_0447 (*mthA*). Additional target genes related to the antibiotic transporter and ribosomal component were selected based on previous review articles [3,49], namely, *dgeo*_1583 (*trkH*) for potassium uptake, *dgeo*_0915 (*nuoG*, a gene of the *nuo* operon) for antibiotic uptake, *dgeo*_0534 (*znuA*, encodes a high-affinity zinc ABC transporter), and *dgeo*_1841 (*rpsD*, encodes ribosomal protein S4). Transposition detection was performed using PCR with the target gene primer sets, followed by agarose gel electrophoresis to identify larger PCR products (Appendix A). After PCR fragments were sequenced, the detected IS element was determined using ISFinder (http://isfinder.biotoul.fr) to search the bacterial IS distribution database [50].

### 4.6. DNA Sequence Analysis 

The PCR products of target genes were amplified and purified using a PCR product purification kit from Bioneer (Seoul, Korea). Appendix A reveals the primer sequences used in this work. DNA sequences were determined by the DNA sequencing facility of Macrogen (Seoul, Korea). Nucleotide point mutations and frameshifts were identified by Blast analysis using BioEdit and ClustalW (http://www.genome.jp/tools-bin.clustalw).

### 4.7. Quantitative Real-Time (qRT)-PCR

To determine the target gene expression levels of two translational elongation factor (EF-Tu) genes (*dgeo*_0646 and *dgeo*_1869) and a chemical modifier gene of streptomycin (*dgeo*_1202), specifically in the Δ*dgeo*_2840 strain, we performed qRT-PCR analysis. We prepared samples at two growth phases with an OD_600_ of 2.0 for the early exponential growth phase and 4.0 for the late exponential growth phase among the wild type, Δ*dgeo*_2840, and both SmR mutants of Δ*dgeo*_2840. The grown cells were harvested, washed with 0.9% NaCl solution, and resuspended. The cell wall was disrupted by phenol extraction solution, followed by DNA digestion by DNaseI treatment. Total RNA was extracted using a RiboEx extraction kit from GeneAll (Seoul, Korea) from the wild type and mutants grown to an OD_600_ of 4. The extracted RNA was purified using a Qiagen RNeasy kit (Hilden, Germany). Using 1 µg of RNA in 8 µL of distilled water, cDNA was synthesized using the PrimeScript^TM^ 1st strand cDNA synthesis kit from TaKaRa (Kusatsu, Japan) with 6-mer random primers and the following conditions: 60 °C for 5 min, 4 °C for 3 min, 30 °C for 10 min, 42 °C for 60 min, and finally, 95 °C for 5 min. In the second step, 4 μL of 5× buffer, 4.5 μL of RNase free water, 1 μL of RTase, and 0.5 μL of RNase inhibitor were added. The amount of cDNA was measured using a Nano-Drop spectrophotometer (DeNovix, Wilmington, DE, USA) and normalized. Next, qRT-PCR was performed using each target gene primer set and cDNA with TB Green^®^ Premix Ex Taq^TM^ from TaKaRa (Kusatsu, Japan) and a CFX96^TM^ Optics Module from Bio-Rad (Hercules, CA, USA). Data were normalized using wild-type GAPDH expression as described previously [45]. All statistical analysis was performed using Prism^TM^ ver. 6. The *t*-test was used to identify differences between samples. A *p*-value less than 0.05 was considered significant.

## Figures and Tables

**Figure 1 ijms-23-09764-f001:**
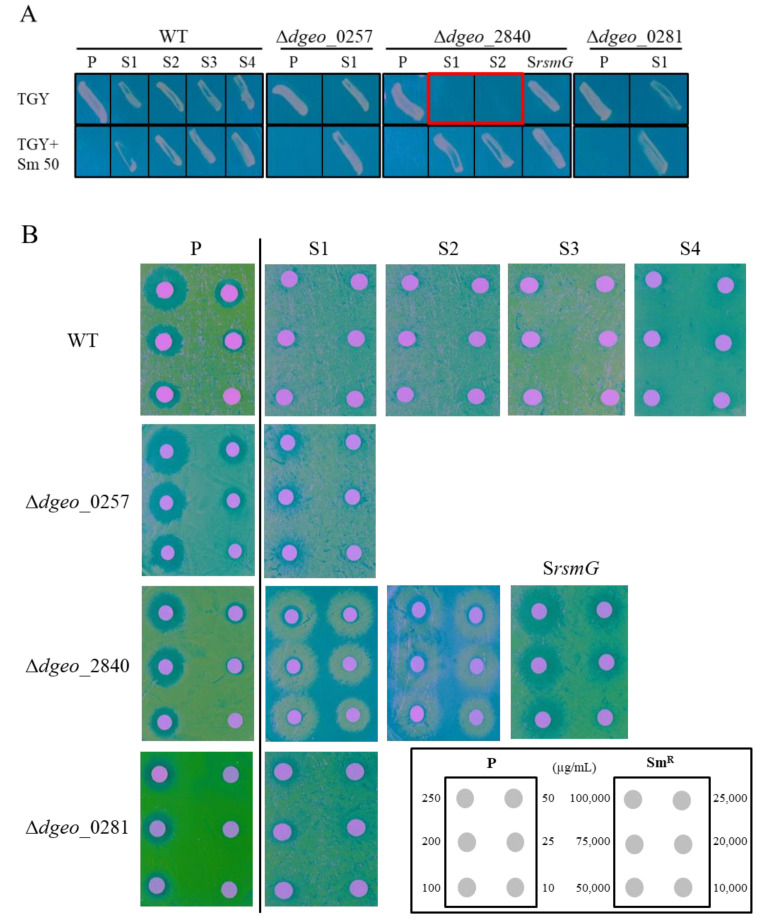
The selection of streptomycin-resistant (SmR) and -dependent (SmD) clones by chronic oxidative stress of wild-type, Δ*dgeo*_0257, Δ*dgeo*_0281, and Δ*dgeo*_2840 parent strains. (**A**) Growth comparison on TGY medium containing 50 µg/mL streptomycin among parent (P) and mutant (S1–S4 and *rsmG*) strains. The red box indicates an Sm-dependent phenotype. (**B**) Minimum inhibitory concentration (MIC) values using the disc diffusion method for all selected mutants within the range of 10 to 250 µg/mL streptomycin for parent strains and 10,000 to 100,000 µg/mL streptomycin for Sm-resistant mutant strains on TGY medium.

**Figure 2 ijms-23-09764-f002:**
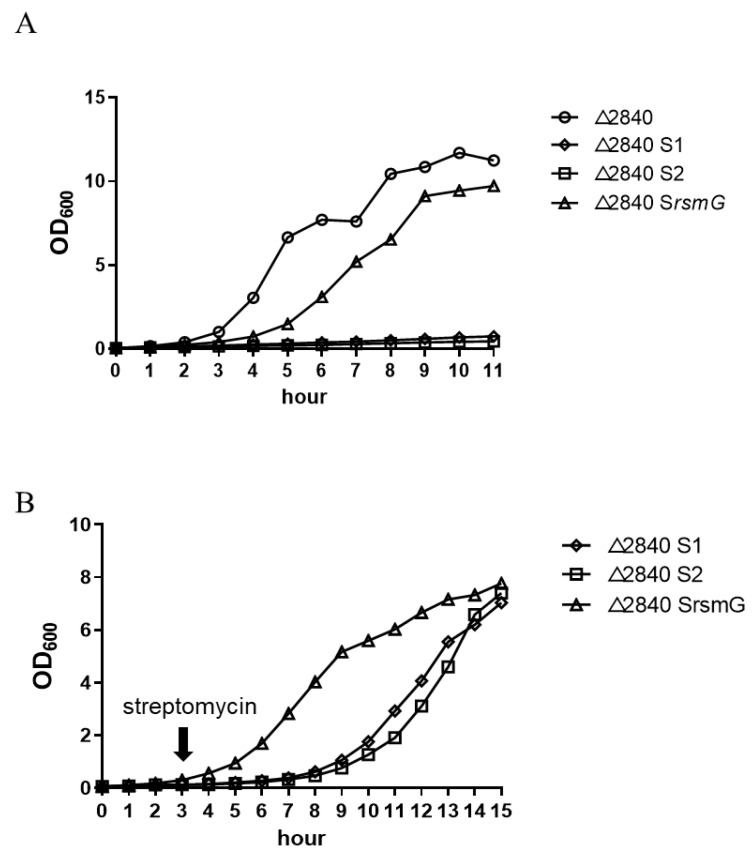
Comparison of growth profiles among Δ*dgeo*_2840 parent and three Sm-resistant mutant strains (**A**) in antibiotic-free TGY medium and the SmD phenotype of Δ*dgeo*_2840 S1 and S2. The arrow indicates the point at which 50 µg/mL streptomycin was added (**B**).

**Figure 3 ijms-23-09764-f003:**
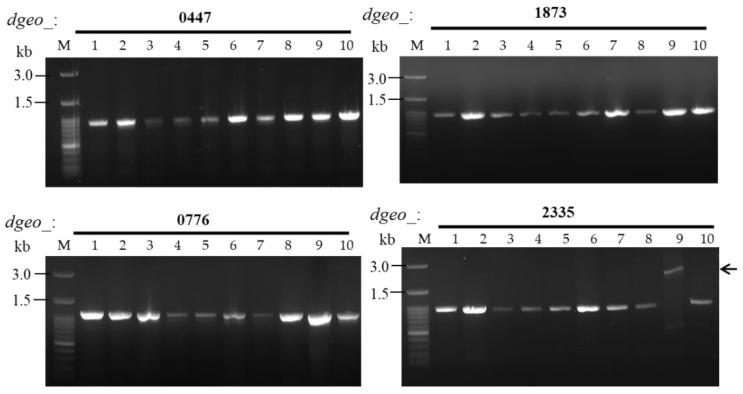
PCR detection of insertion sequence (IS) transposition in four Sm resistance-related genes in wild-type (WT), Δ*dgeo*_0257, Δ*dgeo*_0281, and Δ*dgeo*_2840 mutants. Lanes: M, size marker; 1, WT; 2–5, WT S1–S4; 6, Δ*dgeo*_0257 S1; 7–9, Δ*dgeo*_2840 S1, S2, and S*rsmG*; and 10, Δ*dgeo*_0281 S1. The arrow indicates a larger PCR product caused by IS integration in *dgeo*_2335 (*rsmG*).

**Figure 4 ijms-23-09764-f004:**
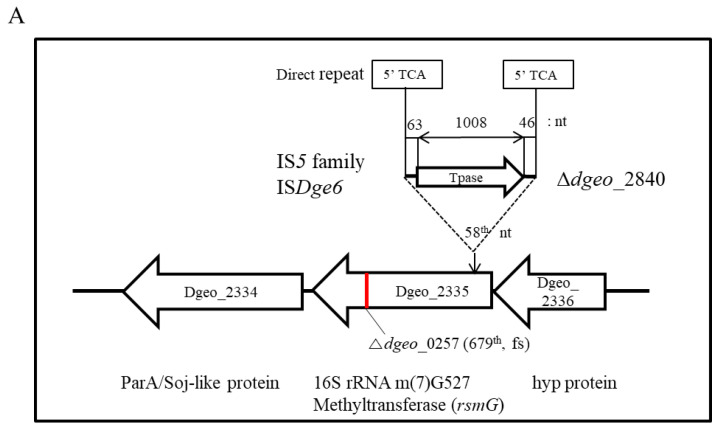
Detection of insertion sequence (IS) transposition and loci of point mutations in the Sm resistance-related genes *rsmG* (*dgeo*_2335) (**A**), *mthA* (*dgeo*_0776) (**B**), and *rpsL* (*dgeo*_1873) (**C**). The IS*Dge6* element of the IS*5* family was integrated at the 58th nucleotide of *rsmG* (**A**). Streptomycin resistance mutants from wild-type, Δ*dgeo*_0257, Δ*dgeo*_0281, and Δ*dgeo*_2840 mutant strains have frameshift or point mutations in three Sm resistance-related genes. All mutations are also summarized in Table 1.

**Figure 5 ijms-23-09764-f005:**
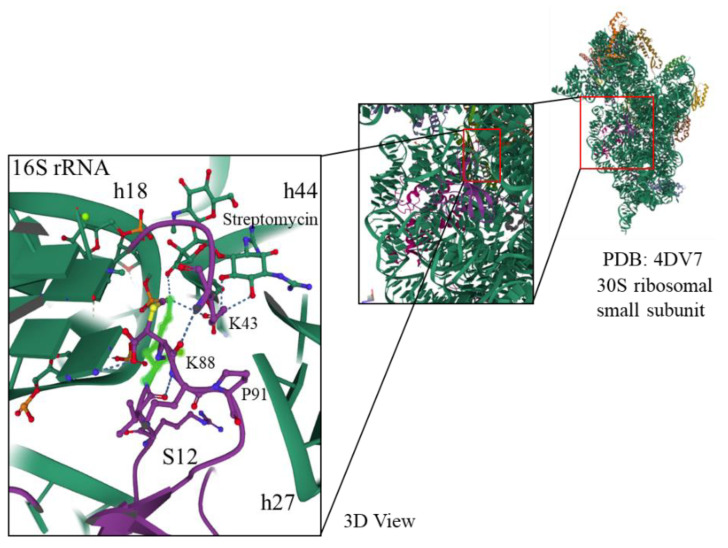
Three-dimensional (3D) view of the decoding center area, including ribosomal proteins S12 and S4 and the three 16S rRNA helices h18, h27, and h44 from the 30S small ribosomal subunit topology of *T. thermophilus* HB8 with streptomycin from the Protein Data Bank (RCSB PDB, www.pdb.org; accession number 4DV7; original contributors: Demirci H, Murphy IVF, Murphy E, Gregory ST, Dahlberg AE, and Jogl G.) [29]. From the deposited structure data, the decoding region was “zoomed-in” using Mol* Viewer [28], and the streptomycin-interacting amino acid residues of ribosomal proteins (violet color) and nucleotides of 16S rRNA helices (green color) were marked.

**Figure 6 ijms-23-09764-f006:**
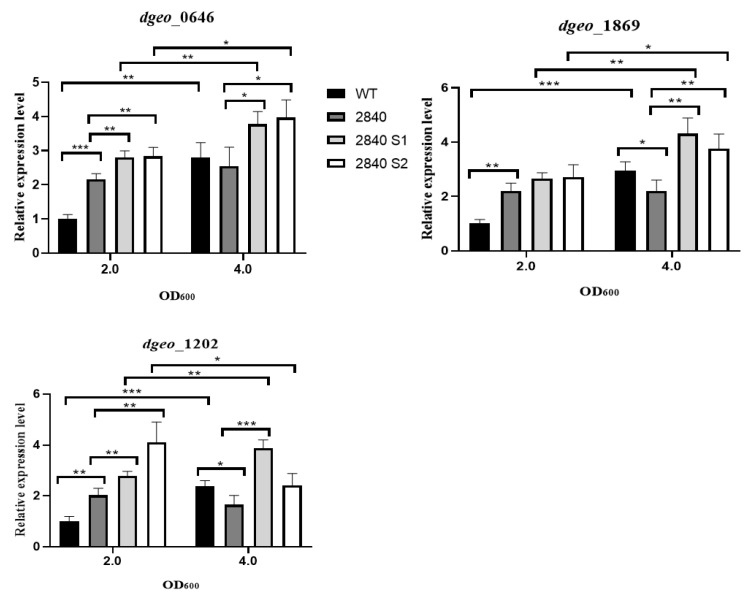
qRT-PCR for Sm-enhanced clone growth of Δ*dgeo*_2840 parent and two SmD strains (S1 and S2) with the wild type at OD_600_ of 2.0 and 4.0. *dgeo*_0646 and *dgeo*_1869 are EF-Tu genes (*tufA* and *tufB*, respectively), and *dgeo*_1202 is a putative *aadA* gene. The statistics were performed with a probability *t*-test in the Prism program with *p* < 0.05 (*), *p* < 0.001 (**), and *p* < 0.0001 (***).

**Table 1 ijms-23-09764-t001:** Detection of mutations in four streptomycin-resistant related genes and an antibiotic uptake gene, their streptomycin MIC values, and their phenotypes.

Strains		Phenotype	MIC (µg/mL)	*dgeo*_1873 (*rpsL*)	*dgeo*_2335 (*rsmG*)	*dgeo_*0776 (*mthA*)	*dgeo*_0447 (*mthA*)	*dgeo*_0915 (*nuoG*)
WT	S1/S3/S4	SmR	50,000	272nd C -> A (P91H)	-	-	-	S3: 1371st C -> A (Arg -> Arg)
	S2	SmR	50,000	272nd C -> A (P91H)	-	84th deletion	-	-
Δ*dgeo*_0257	S1	SmR	20,000	262nd A -> G (K88E)	679th deletionframeshift	-	-	-
Δ*dgeo*_0281	S1	SmR	50,000	272nd C -> A (P91H)	-	102nd G -> C (Arg -> Arg)	627th deletion	1371st C -> A (Arg -> Arg) 1462nd G -> C (Gly -> Arg) 1491st G -> A (Arg -> Arg)
Δ*dgeo*_2840	S1	SmD	25,000	272nd C -> T (P91L)	-	-	-	-
	S2	SmD	25,000	272nd C -> T (P91L)	-	-	-	-
	S*rsmG*	SmR	10,000	-	58th IS*Dge6* transposition	-	-	-

## Data Availability

Not applicable.

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
