# Peer review of "Acquisition of Streptomycin Resistance by Oxidative Stress Induced by Hydrogen Peroxide in Radiation-Resistant Bacterium Deinococcus geothermalis"

_ijms, 2022, doi:10.3390/ijms23179764_

Round 1

Reviewer 1 Report

The authors described well the problems of Streptomycin resistance, giving a good overview of the resistance mechanisms already reported in literature. 

The structure and the methods described in the article are good, but I suggest the authors to consider these suggestion to improve it:

-please provide some further information about the strain considered here, is it a pathogen? why do you decide to highlight the resistance problem in this strain?

-add the numbers to figure and tables to each caption,

-in figure 2B (?), it should be better to add the growth curve at least of the mutant delta2840 rmsG when antibiotic was added to the growth.

-at lines 119 and 190, please indicate the number of Table and figure mentioned

Author Response

Comments and Suggestions for Authors

The authors described well the problems of Streptomycin resistance, giving a good overview of the resistance mechanisms already reported in literature. 

The structure and the methods described in the article are good, but I suggest the authors to consider these suggestion to improve it:

-please provide some further information about the strain considered here, is it a pathogen? why do you decide to highlight the resistance problem in this strain?

> Thank you for your comments. Strain Deinococcus geothermalis is a non-pathogen as a model microorganism for studying molecular mechanisms of DNA repair and oxidative stress responses under higher radiation conditions. Actually, we are interested IS transposition mechanism in radiation-resistant bacterium by the oxidative stress response. For the extension of selectable biomarkers for IS transposition, we choose the Sm-resistant phenotype because this aminoglycoside antibiotic resistance is induced from mutations including point, frameshift, and gene disruption by IS transposition in several selected genes, rsmG, rpsL, and mthA. Thus, we choose antibiotic-resistant, especially streptomycin, to detect DNA sequence profiling of target genes.

We added this reason to select antibiotic-resistance phenotypic changes on 98-99 in the text.

-add the numbers to the figure and tables to each caption,

> Right, our submitted paper was automatically converted to journal form and during this process, there were a lot of miss-typings. Thus, we correctly revised it.

We checked all figures and Tables and revised followed the reviewer’s comments.

-in figure 2B (?), it should be better to add the growth curve at least of the mutant delta2840 rmsG when an antibiotic was added to the growth.

> We added Δdgeo_2840 SrsmG strain with Sm 50 µg/mL condition in Fig. 2B.

-at lines 119 and 190, please indicate the number of Table and figures mentioned

 > Right, our submitted paper was automatically converted to journal form and during this process, there were a lot of miss-typings. Thus, we correctly revised all.

Reviewer 2 Report

In this manuscript, the authors select some streptomycin resistance D. geothermalis strains induced under oxidative stress of H2O2. The mutations occurred at 3 Sm related genes, rsmG, rpsL, and mthA, were examined.

Main questions about the manuscript.

1 Why choose dps mutants (Δdgeo_0257, Δdgeo_0281) and LysR mutant (Δdgeo_2840) for parent strain in this study? Is it important to this study?

2. the frequency (mutation rate) of SmR phenotypic mutation after H2O2 treatment?

3. In Fig 1 and section 4.2, how the SmD phenotype (that is Sm dependent) be determined (or defined) ? It should be described clearly in Section 4.

4. in section 2.1, line 121 and Fig 1A, What is the rsmG-disrupted mutant (Fig 1A as Δdgeo_2840-SrsmG)? Is it obtained from previous reported experiment or this study? To name it as Δdgeo_2840-S3 and then proof it to be a rsmG-disrupted mutant seems to be better. Otherwise describe clearly why it is named as Δdgeo_2840-SrsmG.

5. line 151-154, This sentence describes a speculation, not an experimental derivation. It should be move to discussion.

6. Line 171-172, Is ISDge6 element really ‘specific’ for Δdgeo_2840 strains? What is the evidence?

7. Section 2.5 as well as Fig5, It is not the author’s experimental finding in this study, nor a docking simulation as P91L in RpsL. Even the authors did not refer to the original contributor. All this part (section 2.5) should be move to discussion section. Fig.5 should not be put in this manuscript.

8. In Table 1, It is clearer for reader if lane of MIC & phenotype be moved forward to lane 1 & 2 (after strain name) since they are described earlier.

9. In Table 1, 272nd C-to-A or C->A, instead of 272nd C>A. and so as others in the table.

10. in Section Materials and Methods, the information of chemicals should be provided.

11. Line458-464. It is suggested to be merged to Discussion section. (The conclusion section is not mandatory). Otherwise clearly provide the conclusion.

12. In Abstract section, line 17-24, the Results of this study were not clearly described. One strain with high streptomycin resistance was found with the rsmG insertion by a IS element. Its uniqueness is not clearly proofed. The data of IS element insertion was overemphasized in this section. The disruption of rsmG, whether or not by IS element insertion, conferred to the effect of streptomyces resistance. The possibility that IS element insertion to other related gene and resulting in antibiotic resistance could not be excluded.

Minor comments

13. line 144 and Fig 2, none of the exp data using 100ug/mL, 500ug/mL, 1000ug/mL streptomycin was shown in Fig 2. ‘data not shown’ would be added to maintext. Meanwhile, the experimental condition should be described in Section 4.

14. In Fig 2, using Δdgeo_2840, Δdgeo_2840-S1, Δdgeo_2840-S2…will be better.

15. line 149, ‘exhibited a similar phenotype’ instead of ‘an identical phenotype’.

16. line 162, ‘a longer PCR fragment’ instead of ‘an increased PCR product’

17. Line 199-200, one is a G-to-C substitution…..change, and the other is a …

18. Line 255, what do you mean “early growth phase” ?  

19. Line 271, ‘length of PCR fragment’ instead of ‘PCR product size’.

20. Line 427, After PCR fragment sequenced, the detected IS element ….

21. There are several misending of sentence ending. Such as Line 44, line 198, line 201, line 231,

22. Section subtitle numbered or not?  Line 110, 134, 159, 186, 265, 379, 394, 402, 408, 418, 429, 435. 

23. Figure number? Table number?  Line 127, 156, 174, 180,

Author Response

Comments and Suggestions for Authors

In this manuscript, the authors select some streptomycin resistance D. geothermalis strains induced under oxidative stress of H2O2. The mutations occurred at 3 Sm related genes, rsmG, rpsL, and mthA, were examined.

Main questions about the manuscript.

1 Why choose dps mutants (Δdgeo_0257, Δdgeo_0281) and LysR mutant (Δdgeo_2840) for parent strain in this study? Is it important to this study?

 > Right, we need more explanation for why this work was done in especially dps and LysR-disrupted mutants. As you known that Dps is a chromosome stabilizing protein in the stationary growth phase and stress response conditions. In the case of E. coli, Dps protein is a dominant nucleoid associated protein (NAP) in the stationary growth phase. Actually, dgeo_0257 is a novel Dps candidate in Deinococcus, which is prepared a submission in our group and dgeo_0281 is a major Dps protein that was well characterized. Thus, we are interested in the functional role of Dps on IS transposition under oxidative stress condition (Lee et al., 2019 published) and also in this acquisition of antibiotic resistance. For a similar reason to Dps, a putative LysR family regulator Dgeo_2840 is a candidate signaling transcription regulator for IS transposition under oxidative stress conditions because the LysR family member regulator is a broad spectrum transcriptional regulator under oxidative stress. We reported specialized IS transposition in the dgeo_2840-disrupted mutant (Lee et al., 2020). Unfortunately, still, the signaling pathway for IS transposition is uncharacterized under oxidative stress conditions. Nevertheless, in this work, dgeo_2840-disrupted and complement strains were used antibiotic-resistant phenotypic selection compared with wild-type Deinococcus geothermalis.

  1. the frequency (mutation rate) of SmR phenotypic mutation after H2O2 treatment?

 > Right, it is an important question. We determined the SmR production frequency as 1.36 x 10-7 in the wild-type strain and added SmR frequency on lines 109-111.

  1. In Fig 1 and section 4.2, how the SmD phenotype (that is Sm dependent) be determined (or defined) ? It should be described clearly in Section 4.

 > There are three types of streptomycin-resistant phenotypes such as resistant, dependent, and pseudo-dependent since 1948 (lines 50-52). As two isolated strains from Sm-contained selection medium, especially Δdgeo_2840 based isolated strains exhibit dependent phenotype (Fig. 1A and Fig. 2). Thus, it was indicated in lines 120-121 and in Fig. 1. legend.

  1. in section 2.1, line 121 and Fig 1A, What is the rsmG-disrupted mutant (Fig 1A as Δdgeo_2840-SrsmG)? Is it obtained from previously reported experiment or this study? To name it as Δdgeo_2840-S3 and then proof it to be a rsmG-disrupted mutant seems to be better. Otherwise describe clearly why it is named as Δdgeo_2840-SrsmG.

 > A SmR-selected mutant, SrsmG of Δdgeo_2840, occurred by IS transposition in rsmG gene in this study. Thus, we marked in Fig. 1, 2, and Table 1. In the revised text, it was marked on line 128.

  1. line 151-154, This sentence describes a speculation, not an experimental derivation. It should be move to discussion.

 > Right, this sentence came too early after growth pattern analysis. Thus, this sentence moved to discussion after SmR phenotypes at K43 and K88 of the complementary strain (Fig. S5) on lines 348-350.

  1. Line 171-172, Is ISDge6 element really ‘specific’ for Δdgeo_2840 strains? What is the evidence?

 > In previous our work (Lee et al., 2020), ISDge6 detected the active transposition in Δdgeo_2840 strain under hydrogen peroxide treatment. Recently, we published a methodological paper including DBD plasma and gamma irradiation treatment. At different oxidative stressors, ISDge6 is also broadly transposed at wild-type and some target gene-disrupted strains except Δdgeo_2840 strain, especially gamma irradiation (Ye et al., 2022. J Microbiol Methods 196:106473). Thus, ISDge6 specific transposition in Δdgeo_2840 strain under hydrogen peroxide treatment from present evidence in our data. Actually, the IS transposition specificity needs further analysis because still nobody knows this function in IS transposition mechanism.

  1. Section 2.5 as well as Fig5, It is not the author’s experimental finding in this study, nor a docking simulation as P91L in RpsL. Even the authors did not refer to the original contributor. All this part (section 2.5) should be move to discussion section. Fig.5 should not be put in this manuscript.

 > Right, nevertheless this structure illustration is supported a better understanding of this study, and the results in the process are well fixed to the next qRT-PCR for EF-Tu and others. Thus, we kept Fig. 5 in text and added the original contributors in Fig. 5 legend on lines 244-245.

  1. In Table 1, It is clearer for reader if lane of MIC & phenotype be moved forward to lane 1 & 2 (after strain name) since they are described earlier.

 > Thank you for your kind suggestion. Revised all.

  1. In Table 1, 272ndC-to-A or C->A, instead of 272ndC>A. and so as others in the table.

 > Thanks for the kind comments. We revised it all in Table 1.

  1. in Section Materials and Methods, the information of chemicals should be provided.

 > We provided the used chemicals including antibiotics at lines 402-411.

  1. Line458-464. It is suggested to be merged to Discussion section. (The conclusion section is not mandatory). Otherwise clearly provide the conclusion.

 > Right, the conclusion section was deleted in the revised text.

  1. In Abstract section, line 17-24, the Results of this study were not clearly described. One strain with high streptomycin resistance was found with the rsmG insertion by a IS element. Its uniqueness is not clearly proofed. The data of IS element insertion was overemphasized in this section. The disruption of rsmG, whether or not by IS element insertion, conferred to the effect of streptomyces resistance. The possibility that IS element insertion to other related gene and resulting in antibiotic resistance could not be excluded.

 > Revised clearly in Abstract section on lines 18, 21, and 25-28.

Minor comments

  1. line 144 and Fig 2, none of the exp data using 100ug/mL, 500ug/mL, 1000ug/mL streptomycin was shown in Fig 2. ‘data not shown’ would be added to main text. Meanwhile, the experimental condition should be described in Section 4.

 > added “data not shown” at line 152 and explained the experimental condition on lines 432-434.

  1. In Fig 2, using Δdgeo_2840, Δdgeo_2840-S1, Δdgeo_2840-S2…will be better.

 > Yes, but it doesn’t mean much.

  1. line 149, ‘exhibited a similar phenotype’ instead of ‘an identical phenotype’.

 > Thank you. Revised it.

  1. line 162, ‘a longer PCR fragment’ instead of ‘an increased PCR product’

 > Thank you. Revised it.

  1. Line 199-200, one is a G-to-C substitution…..change, and the other is a …

 > Thank you. Revised it.

  1. Line 255, what do you mean “early growth phase” ?  

 > added “OD600 of 2.0” at line 259. In Materials and methods section, explained the differential growth phases on lines 457-458.

  1. Line 271, ‘length of PCR fragment’ instead of ‘PCR product size’.

 > Thank you. Revised it.

  1. Line 427, After PCR fragment sequenced, the detected IS element ….

 > Thank you. Revised it.

  1. There are several misending of sentence ending. Such as Line 44, line 198, line 201, line 231,

 > Right, our submitted paper was automatically converted to journal form and during this process, there were a lot of miss-typings. Thus, we correctly revised all.

  1. Section subtitle numbered or not?  Line 110, 134, 159, 186, 265, 379, 394, 402, 408, 418, 429, 435. 

 > Right, our submitted paper was automatically converted to journal form and during this process, there were a lot of mis-typings. Thus, we correctly revised all with subtitle numbers.

  1. Figure number? Table number?  Line 127, 156, 174, 180,

> Right, our submitted paper was automatically converted to journal form and during this process, there were a lot of miss-typings. Thus, we correctly revised all in figures and Table.

Reviewer 3 Report

Opinion on Lee et al., "Acquisition of streptomycin resistance by oxidative stress of hydrogen peroxide in radiation resistant bacterium Deinococcus geothermalis"

The authors isolated SmR and SmD mutants of Deinococcus geothermalis after treatment with H2O2. They used various D. geothermalis strains for this experiment: WT, Deltadgeo_0257, Deltadgeo_2840 and Deltadgeo_0281. They analysed several genes of the mutants by PCR amplification and sequencing, including rpsL, rsmG, two mthA (dgeo_0776 and 0447),  znuA, nuoG, trkH and dgeo_1841. They carried out RT-qPCR analysis of three genes (dgeo_0646, 4869 and 1202) in WT and Deltadgeo_2840 strains and the two SmD mutants of the latter.

My major criticism is that the design of the microbiology experiments does not allow the drawing of any conclusion concerning the effect of the oxidative stress or the genetic background. The number of mutants analysed for each genetic background is too small to see the spectrum of mutations, and identify potential significant differences. To detect the effect of the oxidative stress, a control (untreated) mutant spectrum must be observed, and compared to the spectrum seen for the H2O2-treated cells. Again, the analysis of a much larger count of mutants would be required to identify changes in mutation spectra, and attribute it to the effect of the treatment. There is absolutely no quantitative measurement of mutations in this work. Even when selecting for SmR/SmD mutants, there seems to be no determination of complete viable cell numbers, and number of mutants, that would allow at least a mutant frequency calculation (not to mention a flux-balance analysis to calculate the rate of various types of mutation). For these reasons, the conclusions listed at the end of the Abstract can by no means be supported (i.e. IS insertion-, nucleotide exchange- or frameshift mutations being the results of oxidative treatment).

The authors hand-picked a few mutants, and carried out molecular analysis (listed above). This analysis is correctly described, it cannot be used however to support any of the conclusions.

Perhaps as a consequence of these shortcomings, the Conclusions section is completely arcane, unclear and incoherent. I could not extract any take-home message from those three sentences.

Figure numbers missing. Most section numbers missing. Deletions of sentence termini can be found in the following lines: 44, 119, 157, 176, 191, 198, 201, 231.

Author Response

My major criticism is that the design of the microbiology experiments does not allow the drawing of any conclusion concerning the effect of the oxidative stress or the genetic background. The number of mutants analysed for each genetic background is too small to see the spectrum of mutations, and identify potential significant differences. To detect the effect of the oxidative stress, a control (untreated) mutant spectrum must be observed, and compared to the spectrum seen for the H2O2-treated cells. Again, the analysis of a much larger count of mutants would be required to identify changes in mutation spectra, and attribute it to the effect of the treatment. There is absolutely no quantitative measurement of mutations in this work. Even when selecting for SmR/SmD mutants, there seems to be no determination of complete viable cell numbers, and number of mutants, that would allow at least a mutant frequency calculation (not to mention a flux-balance analysis to calculate the rate of various types of mutation). For these reasons, the conclusions listed at the end of the Abstract can by no means be supported (i.e. IS insertion-, nucleotide exchange- or frameshift mutations being the results of oxidative treatment).

  • Thank you for your sharp criticism. Right, this study is not an anti-oxidative stress response mechanism which is a typical network regulation system in various genes involved. Our observations are re-findings for streptomycin-resistant mechanism, especially alternation of ribosomal target modification by low concentration Sm treatment and active IS transposition in Thermus thermophilus excepting oxidative stress by hydrogen peroxide treatment.
  • Measurement of mutational frequency is important for conceptual and phenomenon. We determined the SmR production frequency as 1.36 x 10-7 in the wild-type strain and added SmR frequency on lines 109-111.
  • Our submitted paper was automatically converted to journal form and during this process, there were a lot of mis-typings and miss-endings. Thus, we correctly revised all.
  • The revised sentences were marked with yellow color in the text.

The authors hand-picked a few mutants, and carried out molecular analysis (listed above). This analysis is correctly described, it cannot be used however to support any of the conclusions.

  • In Abstract section, we described clearly our findings which included an active transposition of IS element into the antibiotic-resistant related gene, in this case, rsmG and several point mutations in involved amino acid residues interacting with streptomycin result in streptomycin-resistant phenotypes, especially at K43, K88, and P91 in RpsL (ribosomal protein S12). In general, the Sm-resistant phenotype is obtained from low concentration Sm treatment in various bacteria. Their mutations occurred at streptomycin interacting residues, in the case of Thermus thermophilus K42 and P90. Interestingly, these Sm-resistant mutations are also exactly identical to oxidative stress by hydrogen peroxide treatment in this study. Of cause, as you know that antibiotics also induced oxidative stress. However, we do not enough know about the mutational selection mechanisms for target amino acid residues and cross-network among oxidative stress, IS transposition, and point mutations in target genes.

Perhaps as a consequence of these shortcomings, the Conclusions section is completely arcane, unclear, and incoherent. I could not extract any take-home message from those three sentences.

  • Right, the conclusion section was deleted in the revised text.

Figure numbers missing. Most section numbers missing. Deletions of sentence termini can be found in the following lines: 44, 119, 157, 176, 191, 198, 201, 231.

> Right, our submitted paper was automatically converted to journal form and during this process, there were a lot of miss-typings. Thus, we correctly revised all.

Round 2

Reviewer 3 Report

Opinion on the revised version of Lee et al., "Acquisition of streptomycin resistance by oxidative stress of hydrogen peroxide in radiation resistant bacterium Deinococcus geothermalis"

The conclusions listed at the end of the Abstract have been somewhat rephrased: "Our findings show that the active transposition of a unique IS element under oxidative stress conditions conferred the effect of antibiotic resistance through disruption of rsmG. Furthermore, chronic oxidative stress by hydrogen peroxide also induced streptomycin resistance causing by point and frameshift mutations at streptomycin-interacting residues such as K43, K88, and P91 in RpsL and the four genes for strep- tomycin resistance." I still do not see any evidence for the causative factor of oxidative stress concerning these mutations. That would require analysis of D. geothermalis SmR and SmD mutants that did not undergo oxidative treatment. This is still completely missing.

The authors added the frequency of SmR mutants emerging upon H2O2 treatment. They note that this is "slightly higher than non-pigment production by IS transposition under H2O2 treatment [27]." There are many problems with this sentence. It is of low information to report a value without its variation.  It is very unscientific to compare two values without statistical analysis. And it is meaningless, since we do not know the value of pigment production-disrupting insertion mutants. The authors do not report it in this paper (and I could not find it in reference 27 either). And even if we knew that value, comparing it to the frequency of ALL SmR mutants of this current work would be unjustified, since in this work, the frequency value comprises all types of mutations (point mutation, small indel, IS insertion). And last but not least, reporting the SmR mutant frequency of the same strains without oxidative stress would also be very important to see that the stressor actually has an effect on the mutant frequency.

Overall, the manuscript describes the analysis of a few D. geothermalis mutants (SmR and SmD). This is probably the first description of such mutants in this species, but (as noted by the authors) similar insertion-mutants in rsmG and P91 (or P90) mutations of rpsL resulting in SmR or SmD phenotype have been described in other species. There is no indication whatsoever why similar mutants in D. geothermalis would be unexpected, or of such high interest that they would justify dedicating an entire paper (in a journal as prestigious as IJMS) describing these.